# Self-Assembled CuCo_2_S_4_ Nanoparticles for Efficient Chemo-Photothermal Therapy of Arterial Inflammation

**DOI:** 10.3390/molecules27238134

**Published:** 2022-11-22

**Authors:** Ran Lu, Wei Wang, Bo Dong, Chao Xu, Bo Li, Yong Sun, Junchao Liu, Biao Hong

**Affiliations:** 1Department of Vascular Surgery, The First Affiliated Hospital of Bengbu Medical College, Bengbu 233004, China; 2Department of Vascular Surgery, Tongren Hospital, Shanghai Jiao Tong University School of Medicine, Shanghai 200336, China; 3Department of Vascular Surgery, Shanghai Ninth People’s Hospital, Shanghai Jiao Tong University School of Medicine, Shanghai 200011, China

**Keywords:** self-assembled CuCo_2_S_4_ nanocrystals, chloroquine, arterial inflammation, photothermal therapy, chemotherapy

## Abstract

Cardiovascular disease caused by atherosclerosis (AS) seriously affects human health. Photothermal therapy (PTT) brings hope to the diagnosis and treatment of AS, with the development of nanotechnology. To improve treatment efficiency, self-assembled CuCo_2_S_4_ nanocrystals (NCs) were developed as a drug-delivery nanocarrier, triggered by near-infrared (NIR) light for efficient chemophotothermal therapy of arterial inflammation. The as-prepared self-assembled CuCo_2_S_4_ NCs exhibited excellent biocompatibility and a very high chloroquine (CL)-loading content. In addition, the self-assembled CuCo_2_S_4_ NCs/CL nanocomposites showed good photothermal performance, due to strong absorption in the NIR region, and the release of CL from the NCs/CL nanocomposites was driven by NIR light. When illuminated by NIR light, both PTT from the NCs and chemotherapy from the CL were simultaneously triggered, resulting in killing macrophages with a synergistic effect. Moreover, chemo-photothermal therapy with CuCo_2_S_4_ NCs/CL nanocomposites showed an effective therapeutic effect for arterial inflammation, in vivo. Our work demonstrated that chemo-photothermal therapy could be a promising strategy for the treatment of arterial inflammation against atherosclerosis.

## 1. Introduction

Cardiovascular disease caused by atherosclerosis (AS) seriously affects human health. Disability and fatality rates are continually increasing, but the current treatment methods exhibit varying numbers of deficiencies, such as severe surgical trauma and poor chemotherapy effect [1]. Atherosclerosis is a chronic inflammatory disease, leading to stenosis and occlusion of the lumen. Its pathological manifestations include macrophage enrichment, foam-cell formation, and endothelial-cell dysfunction. It is an important mechanism for macrophages to regulate the inflammatory response of the blood-vessel wall and the formation of AS plaques by phagocytosing lipids and releasing a large number of inflammatory factors to interact with the blood-vessel wall. It has been demonstrated that macrophages play an important role in the formation, development and instability of AS plaques, and promoting their apoptosis can inhibit plaque progression [2]. In addition, the main clinical treatment of AS is balloon dilation and stent implantation, alleviating the symptoms of ischemia, but at the same time activating macrophages to further aggravate the inflammation of the vessel wall and promote the pathological remodeling of the arterial wall [3,4,5]. Therefore, inhibiting the activation, infiltration and proliferation of macrophages, and promoting their apoptosis may be an effective means to inhibit or reverse the inflammation of AS plaques and tube walls.

The pathogenesis of AS so far includes endothelial-cell dysfunction, lipid deposition, immunity and macrophage phagocytosis of lipids, foam-cell formation, macrophage apoptosis, necrotic lipid-core and plaque formation [6]. Most work believes that in the state of cardiovascular diseases, the inflammatory characteristics of macrophages change, releasing a variety of inflammatory factors to form an inflammatory microenvironment in the arterial plaque, and destroying the endothelial barrier, which are the important causes of AS plaque formation [7]. Vascular injury leads to the exfoliation of endothelial cells, platelet adhesion and aggregation, exposure of subintimal tissues (collagen, fibrous tissue, etc.), smooth-muscle-cell migration and proliferation, local thrombosis, and monocyte infiltration, which are the main pathological changes of AS plaques. Therefore, monocytes/macrophages are the important factors affecting the inflammatory blood-vessel wall, and the promotion of macrophages apoptosis can effectively inhibit the progression of AS plaques.

Photothermal therapy (PTT), as a minimally invasive technology, brings hope to the diagnosis and treatment of AS. The PTT technology uses the photothermal effect from photothermal-conversion material to inhibit the proliferation of cells or to kill the cells in the lesion. The near-infrared laser is an important light source that is widely used in the field of photothermal therapy [8,9]. There are many types of reported photothermal agents, which can be mainly classified into four categories, namely precious metals, organic compounds, carbon materials and semiconductor photothermal-agents [10]. In recent years, we have successfully synthesized semiconductor photothermal-agents for the photothermal therapy of AS. Under the combined action of photothermal agents and near-infrared lasers, the number of macrophages in the process of arterial-wall remodeling is significantly reduced, the inflammation of the tube wall is inhibited, and the purpose of preventing restenosis of the tube wall is achieved [11,12]. In addition, we also prepared magnetic Fe_3_S_4_ nanoparticles for the combined photothermal-magnetothermal treatment of AS, which provides a new strategy for the prevention and treatment of deep AS disease [13]. However, heat treatment can significantly inhibit AS-macrophage enrichment in the early stage, but the inhibitory effect is insufficient in the later stage. Therefore, this treatment method needs to be further optimized to improve its treatment efficiency.

One strategy is to combine photothermal therapy with chemotherapy. Among the reported various kinds of photothermal agents, copper-based compounds have been revealed for their use as photothermal agents, due to their excellent photothermal performance, facile synthesis, low cost, and good photostability. The CuCo_2_S_4_ nanoparticles especially have been reported to be appropriate for use as efficient PTT agents prepared by a facile one-pot hydrothermal method [11,14]. The CuCo_2_S_4_ nanoparticles showed excellent photothermal effect, with a photothermal conversion efficiency of up to 73.4% and the capability for magnetic resonance imaging, due to the unpaired 3d electrons of cobalt [14]. Chloroquine (CL) is a kind of anti-inflammatory drug that has been widely used in the treatment of inflammation-related diseases [15]. In this work, photothermal therapy as a minimally invasive technology, combined with chemotherapy, was induced to avoid the severe trauma of surgery and improve treatment efficiency. Self-assembled CuCo_2_S_4_ nanocrystals (NCs) were successfully prepared using a simple one-pot hydrothermal method, and used as an NIR-triggered drug-delivery nanocarrier. Under the irradiation of the NIR laser, the photothermal effect generated from the CuCo_2_S_4_ NCs can promote the release of CL. Moreover, self-assembled CuCo_2_S_4_ NCs as the NIR-driven drug-delivery system could efficiently bring about macrophage apoptosis both in vitro and in vivo. In addition, chemo-photothermal therapy with CuCo_2_S_4_ NCs/CL nanocomposites showed an effective therapeutic effect for arterial inflammation, in vivo. 

## 2. Results and Discussion

The self-assembled CuCo_2_S_4_ NCs was prepared using a facile one-pot hydrothermal method for PTT and drug delivery. The results of transmission electron microscopy (TEM) indicated that the self-assembled CuCo_2_S_4_ NCs were characterized by a hollow disk-shape with a size of 750nm (Figure 1A). Moreover, the size and unique morphology were further confirmed by SEM (Appendix A). Strong light-absorption within the NIR region is an essential feature of the biomaterials used for NIR-induced PTT. As shown in Figure 1B, the UV-vis-NIR absorbance spectrum of the self-assembled CuCo_2_S_4_ NCs exhibited a strong light-absorption within the range of 780 nm to 1100 nm. Moreover, the self-assembled CuCo_2_S_4_ NCs were detected by powder X-ray diffraction (XRD) to depict the crystallographic structure, and the pattern was well indexed to the phase of cubic spinel CuCo_2_S_4_ (Appendix A). Before the drug-loading procedure, the surface area and pore volume of the self-assembled CuCo_2_S_4_ NCs were measured using the BJH and BET methods, to assess their drug-loading property. The results revealed that the surface area and pore size was 315.08 m^2^ g^−1^ and 13.62 nm, respectively (Figure 1C,D), suggesting that the self-assembled CuCo_2_S_4_ NCs could provide sufficient space for further drug delivery. The encapsulation efficiency and loading content of the CL into the self-assembled CuCo_2_S_4_ NCs were determined to be as high as 81.5% and 23.7% (by weight), respectively.

To construct the relationship between laser irradiation and temperature of the self-assembled CuCo_2_S_4_ NCs/CL nanocomposites, the aqueous dispersions of the materials at the concentrations of 0, 25, 50, and 100 ppm were continually irradiated by an 808 nm wavelength laser. The results showed a positive correlation between the duration of laser irradiation and the temperature of the NCs/CL nanocomposites. Notably, after 5-min laser irradiation, the temperature of the aqueous dispersions with a concentration of 100 ppm reached 60 °C (Figure 2A). Furthermore, under the condition of 5-min laser irradiation, the concentration and temperature of self-assembled CuCo_2_S_4_ NCs exhibited an approximate direct proportional-correlation (Appendix A). The release of CL from the self-assembled CuCo_2_S_4_ NCs/CL nanocomposites at pH 7.4 with or without NIR irradiation was assessed, to determine the drug-release property of the CuCo_2_S_4_ NCs/CL nanocomposites. Compared with no NIR irradiation, CL was released faster from the materials under NIR irradiation (Figure 2B). Ten hours after NIR irradiation, >60% CL was released, whereas approximately only 15% CL was released without NIR irradiation. Therefore, the CuCo_2_S_4_ NCs/CL nanocomposites showed NIR-triggered drug-release properties, and were suitable for subsequent in vitro and in vivo experiments. 

Chronic inflammatory-reaction mediated by macrophages is the central part of the pathological and physiological mechanism of AS [16,17,18,19]. On the one hand, macrophages are able to take up lipids and turn them into foam cells, which constitute the principal component of AS plaque [20]. On the other hand, inflammatory macrophages secrete inflammatory cytokines including IL-1, TNF-α, and IL-6, and exacerbate the inflammatory reaction of the artery wall [21]. Raw264.7, a mouse macrophage cell-line, characterized by significant phagocytosis and pseudopodial movement, was widely used in the research of the phenotype and the function of the macrophage [22]. The results of cellular immunofluorescence revealed that Raw264.7 was highly positive for the macrophage marker CD68 (Appendix A). The uptake of biomaterials by macrophages is the premise of our treatment strategy for AS. To confirm that CuCo_2_S_4_ NCs/CL could be specifically phagocyted by macrophages, Raw264.7 cells were incubated with CuCo_2_S_4_ NCs/CL, and then TEM was used to observe the phagocytosis process. Compared with the blank control (Figure 3A,B), self-assembled CuCo_2_S_4_ NCs/CL were obviously phagocyted by Raw264.7, with no damage to the normal cell structure (Figure 3C,D). To calculate the optimal concentration of CuCo_2_S_4_ NCs/CL for the treatment of AS in vivo, CCK-8 and Calcein AM/PI staining assays were then conducted. In order to avoid cell damage while achieving the best therapeutic effect of the materials, we determined their maximum safe-dose on Raw264.7 cells. According to the results of the CCK-8 assay, when Raw264.7 cells were incubated with CuCo_2_S_4_ NCs/CL at the concentration >120 ppm, cell viability was significantly decreased (Figure 4A). However, the cell viability at the concentrations of CuCo_2_S_4_ NCs/CL ≤ 80 ppm showed no obvious difference. Therefore, 80 ppm was the optimal concentration of CuCo_2_S_4_ NCs/CL, and was used for the following experiments. To explore the effect of CuCo_2_S_4_ NCs/CL on the cell viability of macrophages, Raw264.7 cells incubated with 80 ppm of CL, CuCo_2_S_4_ NCs, CuCo_2_S_4_ NCs/CL were stained with Calcein AM/PI. As shown in Figure 4B, after 808 nm NIR laser irradiation, almost all cells were live, with green fluorescence in the control group, but in the CL and CuCo_2_S_4_ NCs group approximately 40% of cells were dead, with red fluorescence. Notably, after NIR irradiation, >90% of Raw264.7 were dead in the CuCo_2_S_4_ NCs/CL group, indicating that PTT could eliminate inflammatory macrophages. The results of the CCK-8 assay further confirmed the excellent PTT efficacy of CuCo_2_S_4_ NCs/CL, which was consistent with previous conclusions (Figure 4C). Accumulation of inflammatory macrophages in the artery wall causes an abnormal immune response and a disturbance of lipid metabolism, leading to the pathogenesis of AS [23]. The elimination of inflammatory macrophages or inhibition of immune response mediated by macrophages is a promising therapeutic target for AS [24,25,26,27]. The current results demonstrated that self-assembled CuCo_2_S_4_ NCs/CL had the dual effects of drug delivery and PTT, which could be utilized to reduce arterial inflammation and relieve AS.

We used the combination of carotid artery wire-injury and high-fat feeding methods to fully mimic arterial inflammation and AS in ApoE^−/−^ mice, as previously described [28]. Two weeks after the surgery, 100 μL of CL, CuCo_2_S_4_ NCs, or CuCo_2_S_4_ NCs/CL (80 ppm) was locally injected into the inflammation site surrounding the carotid artery, and PBS was used as the control (Appendix A). Twelve hours after injection, an 808 nm NIR laser was applied to the body-surface projection of the carotid arteries for 5 min. Two weeks later, the mice were sacrificed and their blood, carotid arteries and vital organs were harvested. To confirm the efficacy of PTT in vivo, the carotid arteries were made into paraffin slices, and HE and immunofluorescence staining were then performed on these slices. The vascular smooth-muscle-cell (SMCs) marker, α-SMA, was used to indicate SMCs in the media of the carotid artery, and the macrophage marker CD68 was used to label inflammatory macrophages infiltrating the artery walls. As shown in Figure 5A,C, significant macrophage infiltration in the control group was observed, and the number of macrophages in the artery wall was decreased in the CL and CuCo_2_S_4_ NCs group after PTT. Moreover, compared with the previous three groups, the infiltrated macrophages were significantly reduced in the CuCo_2_S_4_ NCs/CL group. The results demonstrated that self-assembled CuCo_2_S_4_ NCs/CL could act as a highly efficient drug carrier and photothermal agent for the alleviation of arterial inflammation.

Atherosclerotic plaques and pathological hyperplasia of the intima and media could lead to thrombosis or stenosis [29]. To evaluate the lumen stenosis and the intima and media thickness of carotid arteries, HE staining and morphological analysis was conducted. The results revealed that, compared with the control, the degree of lumen stenosis and the intima and media thickness were partially decreased in the carotid arteries treated with CL and CuCo_2_S_4_ NCs (Figure 5A,B). However, CuCo_2_S_4_ NCs/CL combined with 808 nm NIR laser irradiation significantly improved lumen area and inhibited intima and media hyperplasia, which could increase blood supply to vital organs, and thus reduce the possibility of a fatal ischemic event resulting from AS. Moreover, significant complications of AS such as bleeding or thrombosis were not found during the whole experimental period. The above results demonstrated that self-assembled CuCo_2_S_4_ NCs/CL with irradiation using the NIR laser showed great potential for use in chemo-photothermal therapy for arterial inflammation of AS.

Good biocompatibility is required for the in vivo application of nanotherapy. To detect any tissue toxicity of self-assembled CuCo_2_S_4_ NCs/CL, HE staining was conducted on the vital organs, including heart, kidney, liver, lung, and spleen. No significant morphological or pathological features were found among the CuCo_2_S_4_ NCs/CL, the CuCo_2_S_4_ NCs, the CL and the control groups (Figure 6A). Moreover, no cell necrosis, tissue fibrosis or inflammation infiltration was observed in these tissues. The renal and liver function of the experimental mice were assessed by the detection of serum aspartate aminotransferase (AST), alanine aminotransferase (ALT), creatinine (CR), and blood urea nitrogen (BUN). No significant difference was observed between these four groups in terms of these blood parameters (Figure 6B). These results demonstrated that the CuCo_2_S_4_ NCs/CL had no organ toxicity and no effect on the normal physiological function of the body. In summary, self-assembled CuCo_2_S_4_ NCs/CL was proven to be an ideal therapeutic agent for the treatment of arterial inflammation and AS, with great efficacy and safety.

## 3. Methods

### 3.1. Materials

Copper source, cobalt source, and thiourea were purchased from Shanghai Aladdin Biochemical Technology Co., Ltd. (Shanghai, China). The Raw264.7 cells were obtained from the ScienCell Research Laboratories (California, USA). Dulbecco’s modified Eagle’s medium (DMEM), fetal bovine serum (FBS), penicillin and streptomycin were purchased from Gibco (New York, NY, USA). Anti-CD68 and anti-α-SMA antibodies were obtained from the Abcam company (Cambridge, UK). 6-dimercapto-2-phenylindole (DAPI) was purchased from DAKO (Santa Clara, CA, USA). CCK-8 and Calcein AM/PI reagents were purchased from Sigma-Aldrich (Shanghai, China) Trading Co., Ltd. 

### 3.2. Synthesis of Self-Assembled CuCo_2_S_4_ NCs/CL Nanocomposites

A total of 1.5 mmol of thiourea, 0.5 mmol of CoCl_2_•6H_2_O, 0.25 mmol of CuCl_2_•2H_2_O, and 1 g of poly (vinyl pyrrolidone) were fully dissolved in pure water (40 mL). The resulting solution was kept at 160 °C for 20 h in a hydrothermal reactor. The NCs were obtained by centrifuge and washing with water. CL loading was achieved by mixing CuCo_2_S_4_ NCs and CL under stirring, overnight. The final products were obtained by centrifuge and washing with water.

### 3.3. Characterization

The shape and size of self-assembled CuCo_2_S_4_ NCs can be detected by a transmission electron microscope (TEM). The absorption spectrum of the self-assembled CuCo_2_S_4_ NCs was measured by the UV-vis spectrophotometer. The phase of self-assembled CuCo_2_S_4_ NCs was detected by an X-ray diffractometer (XRD). The release of CL from the self-assembled CuCo_2_S_4_ NCs with or without 808 nm NIR laser irradiation was studied using a UV-vis spectrophotometer.

### 3.4. CL Release In Vitro

The CL-loaded self-assembled CuCo_2_S_4_ NCs (NCs/CL) were collected using centrifugation, and washed three times with PBS solution to remove the unbound CL at different time-points. All the supernatant solution was collected together and measured, using a UV-Vis spectrophotometer, to calculate the amount of CL payload in the self-assembled CuCo_2_S_4_ NCs. The encapsulation efficiency of the self-assembled CuCo_2_S_4_ NCs = (weight of CL loaded into the self-assembled CuCo_2_S_4_ NCs)/(initial weight of CL). The loading content of the self-assembled CuCo_2_S_4_ NCs = (weight of CL loaded into the self-assembled CuCo_2_S_4_ NCs)/(weight of the self-assembled CuCo_2_S_4_ NCs + CL loaded into the self-assembled CuCo_2_S_4_ NCs).The release of CL from the self-assembled CuCo_2_S_4_ nanomaterials with or without 808 nm NIR laser irradiation was studied. The NCs/CL was dispersed in PBS solution at pH 7.4. The laser-triggered drug-release experiments were performed by irradiating the liquid dispersion for 5 min under stirring. The liquid dispersion was then centrifuged and the supernatant was collected. The amount of released CL in the supernatant was determined by using a UV-Vis-NIR spectrophotometer. At certain time intervals, the above operation was repeated.

### 3.5. Cell Culture and TEM Detection

Raw264.7 cells were cultured in DMEM with 4.5 g/L of glucose supplemented with 10% FBS and 1% penicillin and streptomycin at 37 °C and 5% CO_2_. Raw264.7 cells were identified by immunofluorescent staining with the macrophage-specific antibody CD68. After being cocultured with or without CuCo_2_S_4_ NCs/CL (80 μg mL^−1^) for 12 h, the Raw264.7 cells were collected and then observed under TEM [30].

### 3.6. Cell Viability

The cytotoxicity of CuCo_2_S_4_ NCs/CL was firstly assessed in vitro. Raw264.7 macrophages were seeded at a density of 1 × 10^5^ on a 96-well plate, and cultured for 12 h. The cells were then incubated with various concentrations of CuCo_2_S_4_ NCs/CL (0, 20, 40, 80, 120, 200, 400, 800 μg mL^−1^) for 24 h, and a CCK-8 cell proliferation assay was performed to evaluate the cell viability, as described in a previous study [12]. The maximum safe concentration was considered as the optimal concentration, and used for the following studies. Next, the Raw264.7 cells were co-cultured with PBS and 80 μg mL^−1^ of CL, CuCo_2_S_4_ NCs, and CuCo_2_S_4_ NCs/CL for 12 h, followed with 808 nm NIR laser irradiation with a power density of 0.56 W cm^−2^ for 5 min. After washing with PBS, the cells were incubated with CCK-8 agent for 1 h, and the absorbance at 450 nm wavelength was detected. In another set of experiments, Raw264.7 cells were stained with calcein AM/PI and observed under an immunofluorescence microscope (LSM 510 META, Carl-Zeiss-Strasse, Germany).

### 3.7. Animal Model and In Vivo Nanotherapy

All animal experiments were approved by the Animal Ethics Committee of the First Affiliated Hospital of Bengbu Medical College (No. ChiCTR2000040024). Apolipoprotein E knockout (ApoE^−/−^) mice (eight-week-old, male) were fed with a high-fat diet (0.15% cholesterol, 21% fat). The mice were acclimatized to the new environment for 1 week before surgery. The carotid artery wire-injury method was used to induce atherogenesis [28]. Two weeks after the surgery, PBS, CL, CuCo_2_S_4_ NCs, or CuCo_2_S_4_ NCs/CL dissolved in PBS (100 μL, 80 μg mL^−1^) were subcutaneously injected around the carotid artery of the mice (*n* = 8 for each group). Twelve hours after the injection, an 808 nm NIR laser with a power density of 0.56 W cm^−2^ was applied at the injection site for 5 min, as previously described [11]. 

### 3.8. Histological and Blood Analysis

Two weeks after PTT, the mice were euthanized with a pentobarbital overdose and their blood, carotid arteries and vital organs were harvested. Frozen sections of the carotid arteries and paraffin-embedded sections of both carotid arteries and vital organs were prepared. For immunofluorescence analysis, macrophages, smooth muscle cells (SMC) and nuclei were stained with CD68, α-SMA, and DAPI, respectively. The paraffin-embedded sections of the carotid arteries were stained with hematoxylin/eosin (HE). The thickness of the carotid intima and media were analyzed using ImageJ Pro Plus software (Media Cybernetics, Rockville, MD, USA). Furthermore, the sections of heart, kidney, liver, lung, and spleen from the mice were stained with HE to assess the tissue toxicity of CuCo_2_S_4_ NCs/CL. In addition, the blood samples were obtained for liver- and kidney-function detection.

### 3.9. Statistics

All quantitative data from the experiments were reported in the form of mean ± standard deviation (SD). The Student’s *t*-test or a one-way analysis of variance (ANOVA) was applied to determine the statistical significance (*p* < 0.05) of the data. All experiments were repeated three times in every group.

## 4. Conclusions

In conclusion, we developed self-assembled CuCo_2_S_4_ nanocrystals (NCs) as a drug-delivery nanocarrier triggered by near-infrared (NIR) light for the efficient chemophotothermal therapy of arterial inflammation. The as-prepared self-assembled CuCo_2_S_4_ NCs exhibited a high chloroquine (CL)-loading efficiency, due to the mesoporous structure. In addition, the self-assembled CuCo_2_S_4_ NCs showed good photothermal performance, resulting from the strong NIR absorption, and the release of CL from the NCs/CL nanocomposites was driven by NIR light. When illuminated by NIR light, both PTT from the NCs and chemotherapy from the CL were simultaneously triggered, resulting in killing macrophages with a synergistic effect. Moreover, chemo-photothermal therapy with CuCo_2_S_4_ NCs/CL nanocomposites showed an effective therapeutic effect for arterial inflammation, in vivo. Our work demonstrated that CuCo_2_S_4_ NCs/CL nanocomposites under the action of the NIR laser could be used in the efficient treatment of arterial inflammation, against AS.

## Figures and Tables

**Figure 1 molecules-27-08134-f001:**
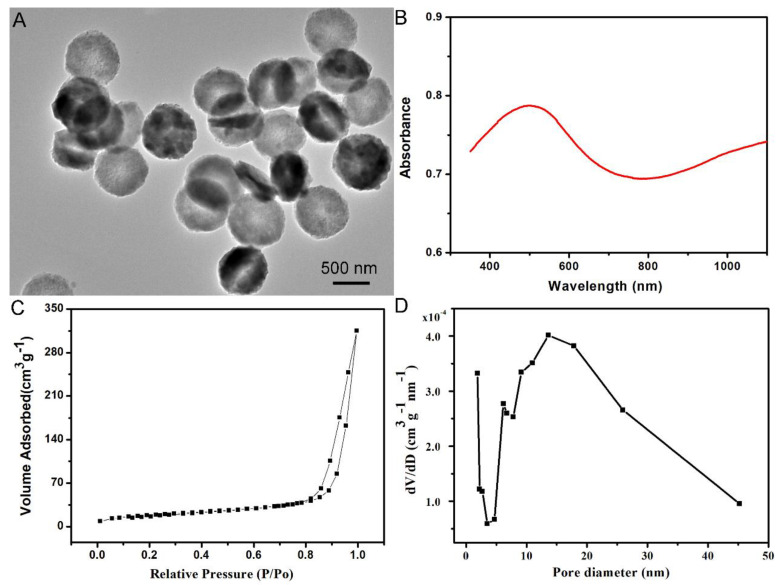
Characterization of self−assembled CuCo_2_S_4_ NCs. (**A**) TEM image of the self−assembled CuCo_2_S_4_ NCs. Scale bar = 500 nm. (**B**) UV−vis−NIR absorption spectrum of the solution of self-assembled CuCo_2_S_4_ NCs at room temperature. (**C**) Surface area of self-assembled CuCo_2_S_4_ NCs. (**D**) Pore−size distributions of self-assembled CuCo_2_S_4_ NCs.

**Figure 2 molecules-27-08134-f002:**
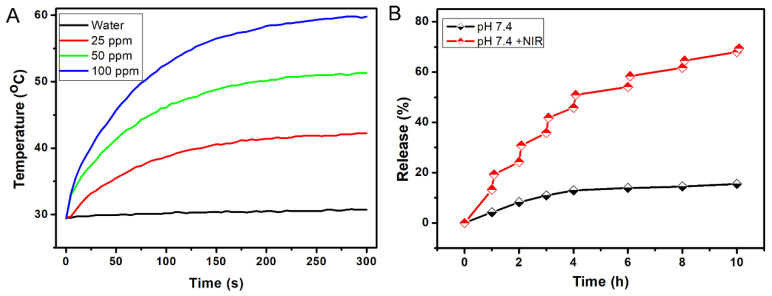
Photothermal and NIR-triggered drug-release properties of self-assembled CuCo_2_S_4_ NCs/CL nanocomposites. (**A**) Temperature-rise curves of self-assembled CuCo_2_S_4_ NCs/CL nanocomposites in aqueous solution at 0, 25, 50, and 100 ppm under the irradiation of the 808 nm NIR laser. (**B**) Drug-release curves of CL from self-assembled CuCo_2_S_4_ NCs/CL nanocomposites in PBS (pH 7.4) with or without 808 nm NIR laser irradiation (0.56 W cm^−2^) at room temperature.

**Figure 3 molecules-27-08134-f003:**
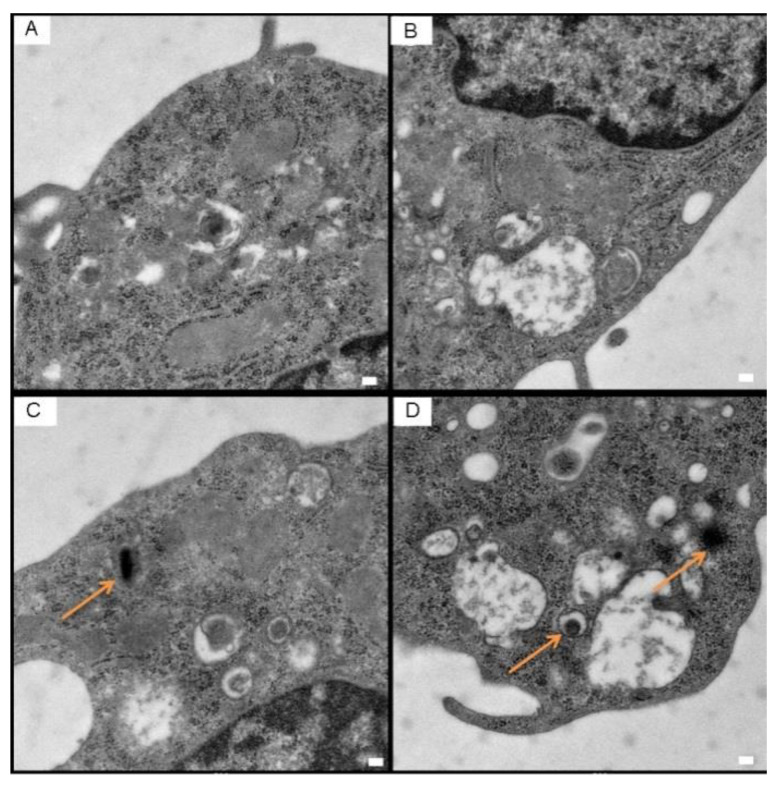
TEM images of Raw264.7 cells incubated with or without CuCo_2_S_4_ NCs/CL. (**A**,**B**) Normal macrophages without CuCo_2_S_4_ NCs/CL. (**C**,**D**) Macrophages incubated with CuCo_2_S_4_ NCs/CL. The phagocyted CuCo_2_S_4_ NCs/CL is marked with arrows. All scale bars = 500 nm.

**Figure 4 molecules-27-08134-f004:**
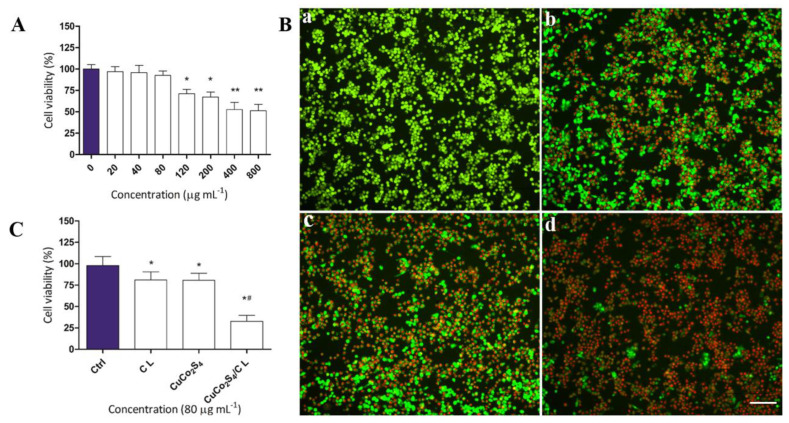
Cytotoxicity and PTT effect of CuCo_2_S_4_ NCs/CL on Raw264.7 cells. (**A**) Relative cell−viability of Raw264.7 cells cultured with the indicated concentrations of CuCo_2_S_4_ NCs/CL for 12 h. *: *p* < 0.05, **: *p* < 0.01. (**B**) Representative images of dead (red, Calcein AM)/live (green, PI) Raw264.7 cells cultured with (**a**) PBS and 80 ppm of (**b**) CuCo_2_S_4_ NCs, (**c**) CL, (**d**) CuCo_2_S_4_ NCs/CL, then irradiated using the 808 nm NIR laser with a power density of 0.56 W cm^−2^. Scale bar = 100 μm. (**C**) Cell viability of Raw264.7 cells cocultured with PBS (Control) and 80 ppm of CL, CuCo_2_S_4_ NCs, CuCo_2_S_4_ NCs/CL followed by the 808nm NIR laser irradiation. *: *p* < 0.05 vs. control, ^#^: *p* < 0.05 vs. CL.

**Figure 5 molecules-27-08134-f005:**
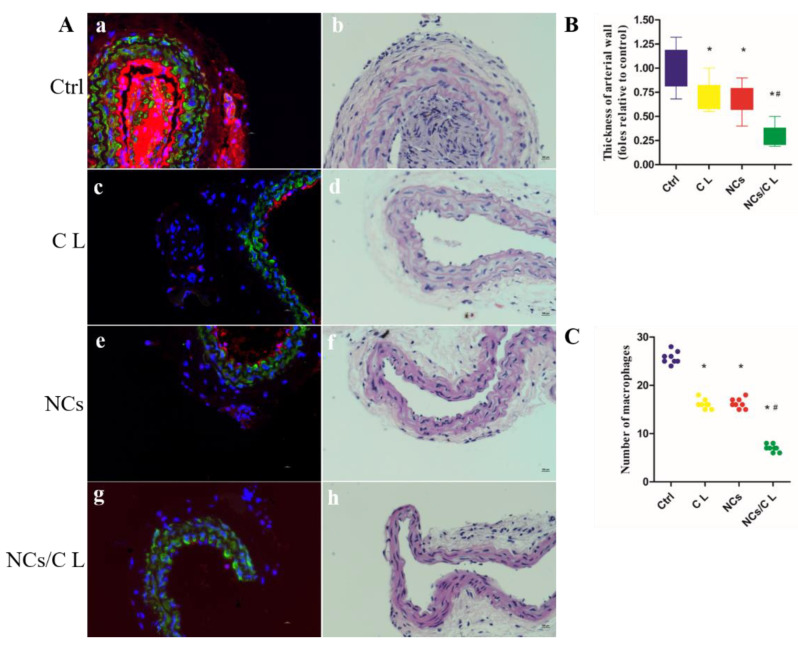
Histological analysis of carotid atherosclerosis in mice. (**A**) Confocal images of CD68 (red) and SMA (green) co-staining of carotid artery treated with PBS (control, (**a**)), CL (**c**), CuCo_2_S_4_ NCs (**e**), and CuCo_2_S_4_ NCs/CL (**g**), followed by NIR laser irradiation, respectively. (**b**,**d**,**f**,**h**) are the representative HE images treated with PBS (control), CL, CuCo_2_S_4_ NCs, and CuCo_2_S_4_ NCs/CL, followed by NIR laser irradiation, respectively. Scale bar = 100 μm. (**B**) Quantification of thickness of arterial wall. (**C**) Quantification of number of macrophages. *: *p* < 0.05 vs. control, ^#^: *p* < 0.05 vs. CL.

**Figure 6 molecules-27-08134-f006:**
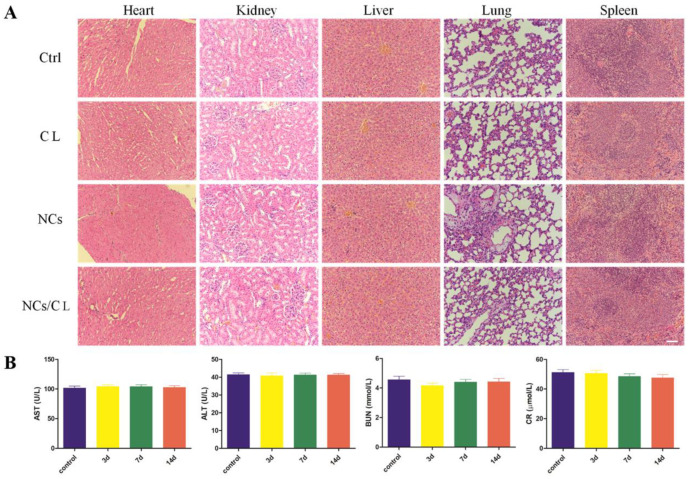
Biocompatibility of self-assembled CuCo_2_S_4_ NCs/CL. (**A**) Representative images of HE staining of the heart, kidney, liver, lung, and spleen from ApoE^−/−^ mice treated with PBS, CL, CuCo_2_S_4_ NCs, and CuCo_2_S_4_ NCs/CL followed by PTT. Scale bar = 100 μm. (**B**) Blood biochemical results of the mice from PBS as a control group and CuCo_2_S_4_ NCs/CL group at different time-points (3 days, 7 days, 14 days) after injection, including AST, ALT, BUN, and CR.

## Data Availability

The data presented in this study are available in Appendix A.

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
