# Peer review of "Self-Assembled CuCo2S4 Nanoparticles for Efficient Chemo-Photothermal Therapy of Arterial Inflammation"

_molecules, 2022, doi:10.3390/molecules27238134_

Round 1

Reviewer 1 Report

The manuscript presented the synthesis of CuCo2S4/CL nanocomposites and evaluated their performance in the photothermal treatment of Raw264.7 cells. The work can be published potentially if the following questions can be addressed adequately.

1.      There are a quite few inconsistencies in the manuscript about the materials. The title stated’ Self-assembled CuCo2S4 nanoparticles for chemo-photothermal therapy’ while the results showed CuCo2S4/CL nanocomposites played a key role. Also, for the results and discussion of Figures 1 and 2, the authors need to clarify whether the materials are CuCo2S4 nanoparticles only or CuCo2S4/CL nanocomposites.

2.      In Figure 1 TEM images, the distributions of alloys in each particle are different. Some particles are light while some particles are dark.

3.      The concentration of CL used in the work is required in the experimental section. Also, the CL loading efficiency needs to be specified.

4.      How was CL release efficiency measured? The information is lacking in the manuscript. Also, error bars are missing in Figure 2B.

5.      The clarification and presentation of figure captions require significant improvement. For example, in figure 4C, it is not clear what control means. In figure 6B, it is hard to tell which figure is for PBS, CL and other conditions, respectively.

6.      The state-of-art of CuCo2S4 preparation and applications should be articulated in the introduction section.

Reviewer 2 Report

To improve the treatment efficiency, self-assembled CuCo2S4 nanocrystals (NCs) were developed as a drug delivery nancarrier triggered by near-infrared (NIR) light for efficient chemophotothermal therapy of arterial inflammation. The as-prepared self-assembled CuCo2S4 NCs exhibited excellent biocompatibility and a very high chloroquine (CL) loading content. Also, the self-assembled CuCo2S4 NCs showed good photothermal performance due to the strong absorption in the NIR region, and the release of CL from the NCs/CL nanocomposites could be driven by NIR light. When illuminated by NIR light, both PTT from NCs and chemotherapy from CL were simultaneously triggered, resulting in killing macrophages with synergistic effect. Moreover, chemo-photothermal therapy with CuCo2S4 NCs/CL nanocomposites showed an effective therapeutic effect for arterial inflammation in vivo. Our work demonstrated that chemo-photothermal therapy could be a promising strategy for the treatment of arterial inflammation against atherosclerosis. This manuscript is well written and contributes significantly to the subject and could be accepted as it is. 

Author Response

Thanks for the reviewer's recognition of our work.

Reviewer 3 Report

The manuscript titled Self-assembled CuCo2S4 nanoparticles for efficient chemo-photothermal therapy of arterial inflammation is well described and the results and discussion were emphasizing the importance of chemo-photothermal therapy in atherosclerosis

The manuscript need minor correction and modification as stated below

 In the introduction, it is stated about the current treatment has more or less deficiencies…….Please describe those deficiencies here, and discuss how your research overcome those.

Copper source, cobalt source, and thiourea were purchased from Aladdin…….provide the manufacturer details

What is the rationale for selecting 800 μg mL−1) as high dose in vitro

Provide the Ethical approval number by Animal Ethics Committee

What is the rationale for selecting only male mice

Describe the method of animal sacrifice.

Statistical analysis methods were not described in the manuscript

What is yellow arrows in figure 3. Make it clear in legend

Is it possible to provide quantitative data of figure 4c?

None of the immune/histo slides has scale bar?

In figure 6, use arrows and make it clear which area of the organ is being focused
